# Niclosamide nanoparticles as a novel adjuvant reverse colistin resistance via multiple mechanisms against multidrug-resistant *Salmonella* infections

Kaifang Yi,[1,2] Peiyi Liu,[1,2] Mengyao Zhang,[1,2] Qiange Liu,[1,2] Mengjing Feng,[1,2] Zibo Li,[3] Dandan He,[1,2] Li Yuan,[1,2] Xiaoyuan Ma,[1,2] Gongzheng Hu[1,2]

**ABSTRACT** The mobile colistin resistance (*mcr*) mechanism enables rapid horizontal transfer of resistance genes across food, animals, and humans, driving significant resistance in *mcr*-carrying bacteria. While numerous adjuvants can reverse colistin resistance, research on their dose–response relationships remains limited, and most suffer from poor solubility, low bioavailability, and safety issues, hindering clinical use. The dose–response analysis showed that niclosamide could achieve a high reversal efficiency within a relatively low concentration range. However, as the concentration of niclosamide increased, the ability to reverse colistin resistance remained unchanged, and the reversal efficiency gradually decreased. Mechanistic analyses reveal that its synergistic antibacterial effect with colistin involves disrupting bacterial membrane permeability, dissipating proton motive force, and inhibiting efflux pumps, leading to membrane damage, cytoplasmic leakage, ATP depletion, and accelerated reactive oxygen species-mediated oxidative damage, ultimately resulting in the death of bacterial cells. A niclosamide nanodelivery system (niclosamide-loaded mPEG-PLGA nanoparticles [NCL@mPEG-PLGA-NPs]) was developed to enhance bioavailability, significantly boosting colistin's efficacy against *Salmonella in vitro* and *in vivo*. The in-depth study of the dose–response relationship of adjuvants in reversing colistin resistance and the establishment of the niclosamide nanodrug delivery system will lay a scientific foundation for the clinical application of colistin adjuvants and the development of suitable drug delivery systems.

**IMPORTANCE** Colistin is used as a last resort for many infections caused by multi-drug-resistant gram-negative bacteria, but colistin-resistant strains are on the rise. Studies have found that the combination of niclosamide and colistin exhibits significant synergistic antibacterial effects. Dose–response analysis shows that niclosamide has extremely high resistance reversal efficiency within a relatively low concentration range. The development of the new dosage form of NCL@mPEG-PLGA-NPs will lay a scientific foundation for the clinical application of colistin adjuvants and the development of drug delivery systems.

**KEYWORDS** colistin, niclosamide, colistin resistance, *Salmonella*, dose–response relationship, nanoparticles

C olistin (COL) is the last-resort antibiotic for treatment of life-threatening multidrug-resistant gram-negative bacterial infections (1). However, since the occurrence of a set of transmissible resistance genes mediated by the mobile colistin resistance (*mcr*) on plasmids, the antibacterial effectiveness of colistin has recently been greatly challenged (2–4). Notably, unlike traditional chromosomally carried resistance pathways,

Address correspondence to Xiaoyuan Ma, lisamxy@126.com, or Gongzheng Hu, yaolilab@126.com.

The authors declare no conflict of interest.

See the funding table on p. 19.

the new plasmid-mediated *mcr* resistance mechanism allows the rapid horizontal transfer of resistance genes between people, food, animals, and the environment (5, 6). Therefore, there is an urgent need to find new therapies to overcome *mcr*-mediated acquired colistin resistance in *Salmonella*. Finding antibacterial adjuvants is an alternative promising strategy for fighting colistin-resistant bacteria by targeting resistance mechanisms, compared to the high costs and unpredictability of developing new antimicrobial agents (7). For instance, combining colistin with additional antibiotics or drugs, such as clarithromycin, rifampicin, auranofin, and artemisinin, has been investigated to increase its antibacterial efficacy against colistin-resistant bacteria (8). Nevertheless, in recent years, more than 80 kinds of adjuvants that can reverse colistin resistance have been reported. However, due to the lack of research on the dose–response relationship between adjuvants and their resistance-reversing effects, as well as issues regarding the effectiveness, safety, specificity, and reasonable dosage of adjuvants, none of them has been successfully applied in clinical settings.

Niclosamide (NCL) was approved by the U.S. Food and Drug Administration (FDA) for use in humans to treat tapeworm infection in 1982 and the World Health Organization's list of essential medicines (9). Niclosamide exhibits antiviral activity against severe acute respiratory syndrome coronavirus and is an effective antibiotic against gram-positive and acid-fast pathogens (e.g., *Staphylococcus aureus*, *Clostridium difficile*, and *Mycobacterium tuberculosis*), as well as against *Helicobacter pylori* (10). Previous studies have found that *in vitro* combination therapy with niclosamide and colistin overcomes colistin resistance in *Escherichia coli*, *Acinetobacter baumannii*, and *Klebsiella pneumoniae* (11, 12). However, there has been no research on the dose–response relationship between niclosamide and its reversal effect on colistin resistance. In addition, the low solubility and low bioavailability of niclosamide also limit its clinical application.

The dose–response relationship of adjuvant reversing resistance and its bioavailability are important factors affecting the clinical application of adjuvants. Therefore, we focus on studying the dose–response of niclosamide in reversing colistin resistance of *Salmonella* isolates and preparing nanoparticles (NPs) to improve its bioavailability. In addition, poly(lactic-co-glycolic acid) (PLGA) and methoxy poly(ethylene glycol) (mPEG) approved by the FDA have been extensively applied to prepare nanoparticles and have shown good biocompatibility *in vivo*, with their degradation products of carbon dioxide and water being highly safe (13). Here, we prepared core-shell nanoparticles (niclosamide-loaded mPEG-PLGA nanoparticle [NCL@mPEG-PLGA-NPs]) by double emulsification using the amphiphilic copolymer methoxy poly(ethylene glycol)-poly(lactide-co-glycolide) (mPEG-PLGA) to improve the bioavailability of niclosamide and enhance the therapeutic effect against colistin-resistant *Salmonella* infection.

In this study, we found that niclosamide could significantly reverse the mcr-mediated resistance to colistin both *in vivo* and *in vitro* through different mechanisms. The dose–response analysis demonstrated that niclosamide had an extremely high resistance reversal efficiency within a relatively low concentration range. The study of the dose–response relationship provides practical references for determining the optimal formulation or dose ratio of colistin:niclosamide, and the development of the new dosage form of NCL@mPEG-PLGA-NPs will lay the foundation for the clinical application of colistin adjuvants.

## MATERIALS AND METHODS

### Bacterial strains and growth conditions

The clinical *Salmonella* isolates were acquired from samples from a chicken farm and chicken slaughterhouses in Henan Province, China, including 10 *mcr*-1 positive and 9 *mcr*-1 negative strains (Table S1 and Fig. S1). All strain identifications were confirmed by PCR analysis and 16S rRNA sequencing, along with matrix-assisted laser desorption/ionization time-of-flight mass spectrometry detection (AXIMA Performance; Shimadzu Corporation, Kyoto, Japan) (14).

## Antimicrobial susceptibility testing

The minimum inhibitory concentration (MIC) tests were performed in triplicate using broth microdilution in accordance with Clinical and Laboratory Standards Institute recommendations (15). Briefly, drugs were twofold diluted in Mueller–Hinton broth (MHB) and mixed with an equal volume of bacterial suspensions containing approximately $1 \times 10^5$ CFU/mL in a 96-well microliter plate. After 18 h incubation at 37°C, the MIC values were defined as the lowest concentrations of antibiotics with no visible growth of bacteria.

The minimum resistance reversal concentration (MRC) was defined as the lowest concentration of niclosamide that can inhibit bacterial growth in the presence of colistin at susceptibility breakpoint (2 µg/mL). In brief, the fixed colistin concentration was 2 µg/mL, and the twofold broth microdilution method was used to determine the minimal niclosamide concentration with no visible growth of bacteria after 18 h incubation at 37°C.

## Checkerboard assay

The assay was performed on a 96-well plate as previously described (16). The antibiotic of interest was twofold serially diluted along the x-axis, whereas niclosamide was twofold serially diluted along the y-axis to create a matrix, where each well consists of a combination of both agents at different concentrations. Bacterial cultures grown overnight were then diluted in saline to $OD_{600} = 0.5$, followed by 1:1,000 further dilution in MHB and inoculation on each well to achieve a final concentration of approximately $5 \times 10^5$ CFU/mL. Wells comprising MHB with or without bacterial cells were used as positive or negative controls, respectively. The 96-well plates were then incubated at 37°C for 18 h.

The synergistic effect was demonstrated as the fractional inhibitory concentration index (FICI):

FICI = (MIC of colistin in combination / MIC of colistin) + (MIC of NCL in combination / MIC of NCL).

FICI was interpreted as follows: synergistic, FICI < 0.5; additive, 0.5 < FICI < 1; indifferent, 1 < FICI < 2; and antagonistic, FICI > 2.

## Time–kill kinetic assay

Time–kill curves of the bacteria were performed in duplicate. *Salmonella* SH05 and 46R culture grown overnight was diluted in Luria–Bertani (LB) medium to $OD_{600} = 1.00$ followed by a further 1:100 dilution in fresh MHB. Bacteria were then treated with niclosamide (1 or 4 µg/mL) and colistin (1/2× MIC) alone or their combination. At the time points 0, 2, 4, 8, 12, and 24 h, 100 µL aliquots were obtained from each tube, resuspended in phosphate-buffered saline (PBS), and serially diluted. Subsequently, the dilutions were spotted on LB agar plates and were incubated for 24 h at 37°C, and after colony counts, the $\log_{10}$ (CFU/mL) of viable cells was determined. Synergy was defined as the reduction of ≥2 $\log_{10}$ CFU/mL with the combination concerning the more active drug (17).

## Resistance development studies

A modified serial passage protocol was developed based on previously described methods (18). The bacterial suspension with an $OD_{600}$ of 0.1 was treated with colistin monotherapy (sub-MICs of colistin) or the combination therapy of niclosamide (1 or 4 µg/mL), and colistin was determined after a 30-day serial passage assay, with no treatment as a control, and samples were incubated for another 18–24 h with shaking at 37°C. The bacterial cells growing at the highest concentration (1/2× MIC) were harvested after 24 h of shaking incubation at 37°C to serve as the working inoculum for the subsequent experiment day. Then, this was repeated for the remaining duration of 30

days of the assay; the MIC values for colistin were recorded and plotted as the MIC fold change compared to the experiment's first day.

## Study on dose–response relationship of reversing colistin resistance

The concentration of niclosamide was fixed at MRC-sub-MICs to explore the best adjuvant dose range for reducing colistin MIC. The niclosamide concentration that maximizes the reversal fold (RF) of colistin resistance is defined as the peak reversal concentration (PRC). The PRC and resistance reversal index (RRI) of niclosamide were determined by the microbroth dilution method, and the dose–response relationship curve was drawn. Among them, the RRI was the RF of colistin resistance divided by the increased fold (IF) of corresponding niclosamide MRC. RF was calculated by dividing the susceptibility breakpoint concentration of colistin (2 µg/mL) by the MIC of colistin in the presence of niclosamide. Similarly, IF in the MRC of niclosamide was calculated by dividing its tested concentration by the MRC of niclosamide (19).

The correlation between the reversal effect and concentration of niclosamide was calculated with the following equation.

$$RRI = RF/IF = \frac{2}{MIC_{(COL)}} / \frac{C_{(NCL)}}{MRC},$$

where "2" is the susceptibility breakpoint of colistin; $MIC_{(COL)}$ is the MIC of colistin in the presence of a corresponding concentration of niclosamide; $C_{(NCL)}$ is the niclosamide concentration; MRC is the minimum resistance reversal concentration of niclosamide.

RRI was interpreted as follows: high-efficiency reversal, RRI ≥ 1; medium-efficiency reversal, 0.5 ≤ RRI < 1; and low-efficiency reversal, RRI < 0.5.

## Fluorescence assay

Pretreatments of biochemical assays were performed using similar protocols as follows. The tested strain (*Salmonella* SH05) was grown overnight at 37°C with shaking at 200 r/min. Then the cultures were washed and suspended with 5 mM 4-(2-hydroxyethyl)-1-piperazineethanesulfonic acid (pH 7.0, plus 5 mM glucose). The bacterial suspension's absorbance at $OD_{600}$ was standardized to 0.5 in the same buffer, and the fluorescent dye was added. After incubation at 37°C for 30 min, an aliquot of 1 mL of bacterial suspension was mixed with niclosamide (1 µg/mL) or colistin (1/2× MIC) alone or in combination. After incubation for 1 h, 200 µL bacterial suspension was added to the 96-well plate white with flat bottom. Subsequently, fluorescence intensity or luminescence was measured by a Spapk 10 M microplate reader (Tecan).

## Outer membrane permeability

1-N-Phenyl-naphthylamine (NPN) (10 µM) assay was used to assess the outer membrane (OM) permeability according to a previous study (20). Fluorescence intensity was measured with the excitation wavelength at 350 nm and emission wavelength at 420 nm by a Spapk 10 M Microplate reader (Tecan).

## Membrane potential assay

3,3-Dipropylthiadicarbocyanine iodide [$DiSC_3(5)$] (0.5 µM) was applied to determine the membrane potential (21). The dissipated membrane potential of *Salmonella* SH05 was measured with an excitation wavelength of 622 nm and an emission wavelength of 670 nm.

## ATP determination

Intracellular ATP levels of tested strains were determined using an Enhanced ATP Assay Kit (Beyotime, China). The luminescence of the supernatant was monitored using the

Spapk 10 M Microplate reader (Tecan). The intracellular ATP levels of the tested strains were calculated based on the luminescence signals.

## Efflux pump activity assay

The accumulation of ethidium bromide (EtBr) (Beyotime) was carried out as previously described (22). The accumulation of EtBr in the cells was monitored with an excitation wavelength of 530 nm and an emission wavelength of 600 nm.

## Total reactive oxygen species measurement

2′,7′-Dichlorodihydrofluorescein diacetate (10 µM) was applied to monitor levels of reactive oxygen species (ROS) in *Salmonella* SH05 following the manufacturer's instruction (Beyotime) (23). Fluorescence intensity was measured with an excitation wavelength of 488 nm and an emission wavelength of 525 nm. The ROS scavenger L-ascorbic acid (L-Aa, 10 mM) was used as a control to neutralize the production of ROS (24).

## Reverse transcription PCR analysis

*Salmonella* SH05 cultured overnight was treated with niclosamide (1 µg/mL) or colistin (1/2× MIC) alone or in combination for 4 h. RNAiso Plus (Vazyme) was used to extract the total RNA, and PrimeScript RT reagent Kit (TaKaRa) was used to reverse transcribe the RNA into cDNA. Before reverse transcription, the extracted RNA concentration was adjusted to be the same. Reverse transcription PCR analysis was performed by 7500 Fast Real-Time PCR System (Applied Biosystem, USA) using the ChamQ Universal SYBR Qpcr Master Mix (Vazyme) with the optimized primers, and 16S rRNA of the strain was used as a reference gene in Table S2. Each quantification was performed in duplicate. The fold changes in gene expression were determined using the $2^{-\Delta\Delta Ct}$ method (25).

## Molecular docking

A molecular docking study was performed to investigate the binding mode of niclosamide to MCR-1 and TolC using AutoDock Vina 1.1.2. The crystal structure of MCR-1 (PDB ID: 5GRR), available at Protein Data Bank, was obtained in PDB format. The structure of niclosamide was obtained from the PubChem Project (PubChem CID: 4477). The number of rotatable keys in the ligand as well as spatial interaction, hydrophobic interaction, and hydrogen-bonding energy are weighting parameters of the scoring function. The docking is evaluated by measuring affinity. The smaller the parameter, the more likely it is that the ligand will bind to the active pocket. The PyMoL molecular graphics system was used for analysis of their modes of interaction with binding site residues (26).

## NCL@mPEG-PLGA-NPs preparation and characterization

NCL@mPEG-PLGA-NPs were prepared using the double emulsion (water-in-oil-in-water [W/O/W]) method. In brief, NCL dissolved in acetone (2 mg/mL, 0.5 mL) was emulsified in 0.5 mL dichloromethane (DCM) containing 25 mg of mPEG-PLGA copolymer via probe sonication (acetone:DCM = 1:1). The water-in-oil emulsion was mixed with 3 mL polyvinyl alcohol (PVA) and resonicated (PVA, 1% wt/vol). The resulting W/O/W emulsion was slowly dropped into PVA (0.5% wt/vol, 20 mL) and stirred for 10 min at room temperature. After vacuum evaporation of the solvent was achieved, the NPs were freeze-dried under vacuum (27). Ultimately, the collected NCL@mPEG-PLGA-NP freeze-dried powders were stored for further experiments.

### *Box–Behnken response surface method*

Based on the single-factor test, three factors with significant effects on preparation of NCL@mPEG-PLGA-NPs were screened: mPEG-PLGA concentration, acetone-to-dichloromethane volume ratio, and PVA concentration. The encapsulation efficiency (EE), drug

loading efficiency (LE), and particle size were used as the response values. The Box–Behnken method, combined with the software Design Expert 11.0, was used to optimize the design. Niclosamide was quantified using UV-vis spectrometry. Each experiment was repeated three times. The encapsulation efficiency and drug loading efficiency of the NPs were calculated by the following equations (28):

$$LE = \frac{\text{Mass of NCL in NPs}}{\text{Mass of NPs}} \times 100\%$$

$$EE = \frac{\text{Mass of NCL in NPs}}{\text{Mass of the feeding NCL}} \times 100\%.$$

The ultra-micromorphology of the samples was determined by a Tecnai 12 transmission electron microscope (Royal Philips, Netherlands). The particle size, polydispersity index (PDI), and zeta potential of the samples were determined by a Zetasizer Nano ZS90 nanoparticle size analyzer (Malvern, UK). A Cary 610/670 microinfrared spectrometer (Agilent Technologies Ltd., Santa Clara, CA, USA) was used to scan in the wave number range of 400–4,000 cm$^{-1}$ to obtain the infrared spectra of each sample.

## Safety assessment

Ten percent mice blood cells were treated with niclosamide (4–64 µg/mL) or NCL@mPEG-PLGA-NPs (16–256 µg/mL) for 1 h. Pure water was used as a positive control. After incubation, the absorbance of supernatant at 540 nm was measured, and the hemolysis rate was calculated by comparing with the positive control. To evaluate the *in vivo* biocompatibility, BALB/c mice (6–8 weeks) were randomly divided into three groups (*n* = 4) and treated with COL + NCL (5 + 5 mg/kg), COL + NCL@mPEG-PLGA-NPs (5 + 20 mg/kg), or COL (5 mg/kg) via intraperitoneal injection, respectively. After 72 h of observation, blood samples were collected from the orbital veins of the mice and centrifuged to obtain serum for biochemical analysis.

## Therapeutic effect of NCL@mPEG-PLGA-NPs in combination with colistin in a murine model of peritoneal sepsis

The mouse peritoneal sepsis model was used in the *in vivo* experiment to assess the effectiveness of niclosamide and colistin for the treatment of *Salmonella* SH05 infection. Male BALB/C mice, 4 weeks old and weighing about 20 g, were purchased from the Huaxing Experimental Animal Center (Zhengzhou, China). All animals were acclimated to the controlled environment for 1 week prior to the experiment.

Forty-eight mice were randomly ascribed to six groups and intraperitoneally infected with a dose of $2.0 \times 10^7$ CFU/mL *Salmonella* SH05 suspension. At 1 h post-infection, mice were treated with a single dose of colistin (5 mg/kg), niclosamide (10 mg/kg) alone, the combination of colistin with niclosamide (5 + 10 mg/kg), the combination of colistin (5 mg/kg) with NCL@mPEG-PLGA-NPs (20 or 40 mg/kg: 20 and 40 mg/kg of NCL@mPEG-PLGA-NPs, which contain equivalent doses of 5.48 and 10.96 mg/kg of niclosamide, respectively) via intraperitoneal injection. Survival rates of treated mice were recorded for 7 days (21).

Sixty mice were randomly ascribed to six groups and infected intraperitoneally with $5.0 \times 10^6$ CFU/mL *Salmonella* SH05. The mice were treated with a single dose of colistin (5 mg/kg), niclosamide alone (10 mg/kg), NCL@mPEG-PLGA-NPs alone (40 mg/kg), combination of colistin with niclosamide (5 + 10 mg/kg), combination of colistin (5 mg/kg) with NCL@mPEG-PLGA-NPs (20 or 40 mg/kg) via intraperitoneal injection at 1 h post-infection. At 48 h post-infection, mice were euthanized by cervical dislocation, and the spleen and liver were removed, weighed, homogenized, and serially diluted in PBS. The bacterial counts in the target tissues of each group were computed and presented as the mean ($\pm$ SD) log$_{10}$ CFU/g tissue.

## Statistical analysis

Statistical analysis was performed using GraphPad Prism 8, Origin2021, and SPSS software. Each experiment included at least three independent biological replicates. All

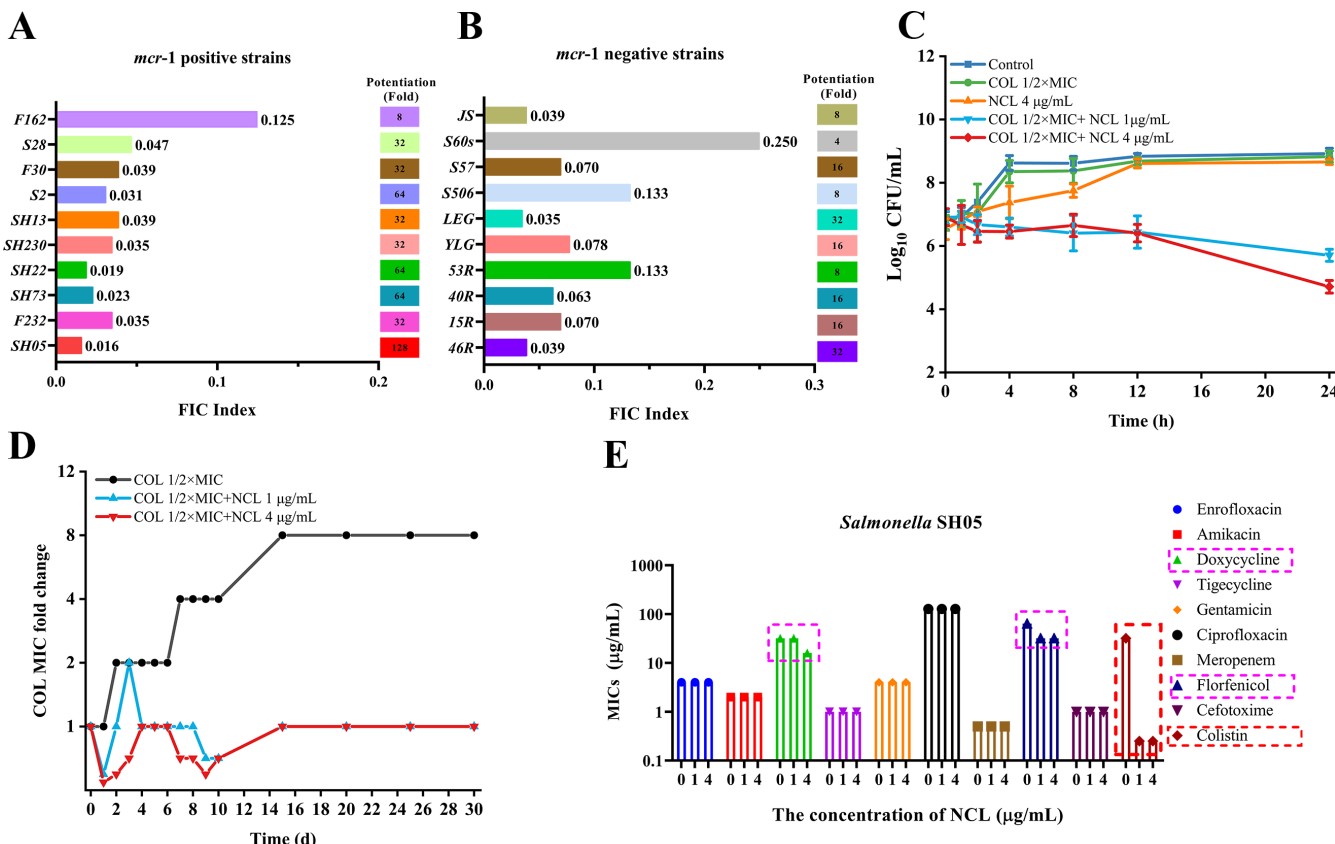

**FIG 1** Niclosamide was an effective adjunct to reverse colistin resistance. (A and B) Synergistic activity of niclosamide with colistin (FICI) against diverse strains of *mcr*-1-positive and *mcr*-1-negative colistin-resistant *Salmonella*. (C) Time–kill curve of *salmonella* SH05 (*mcr*-1 positive) treated with the combination of niclosamide (1 or 4 μg/mL) and colistin (1/2× MIC). All data represent the means from three independent experiments. (D) Emergence of bacterial resistance in *Salmonella* SH05 after 30 serial passages with 1/2× MIC colistin in the presence or absence of niclosamide. (E) Among 10 different antibiotics, only colistin (outlined in red) and niclosamide (1 or 4 μg/mL) significantly exhibited synergistic activity against *Salmonella* SH05. The data represent one of three independent experiments.

data are presented as mean ± SD. Unpaired *t*-tests were used to calculate *P* values (*$P <$ 0.05, **$P < 0.01$, ***$P < 0.001$).

## RESULTS AND DISCUSSION

### Synergistic activity of niclosamide and colistin against colistin-resistant *Salmonella*

*In vitro* activity of niclosamide alone or in combination with colistin was tested against reference and wild-type *Salmonella*. Niclosamide alone showed a high MIC value of 1,024 μg/mL, which revealed that it had no direct antimicrobial activity against *Salmonella*. More importantly, niclosamide was found to significantly synergize with colistin against all the tested resistant strains using a checkerboard assay, with an FICI of 0.016–0.25 (Fig. 1A and B). Interestingly, for resistant strains with or without *mcr*-1, niclosamide potently reversed colistin resistance; MIC of colistin was reduced below the susceptibility breakpoint; and FICI was lower than 0.25 for all tested strains. Notably, when niclosamide was combined with 2 μg/mL colistin, the MRC values of niclosamide ranged from 0.016 to 0.5 μg/mL, accompanied by a 4,096- to 65,536-fold decreases in MIC values of niclosamide (Table 1). Our data confirmed that the low concentrations of niclosamide reversed colistin resistance in all tested strains.

The results of time–kill curve showed that 1 or 4 μg/mL niclosamide, in combination with 1/2× MIC colistin, demonstrated higher synergistic activity at 24 h. The bacterial cell

**TABLE 1** MRC of niclosamide on colistin resistance against colistin-resistant *Salmonella*[a]

| Strains | *mcr*-1 | MICs | | MRCs (µg/mL) | |
|---|---|---|---|---|---|
| | | COL | NCL | 2 µg/mL COL + NCL | NCL MIC fold change |
| 15R | Negative | 16 | 1,024 | 0.0625 | 16,384 |
| 40R | Negative | 16 | 1,024 | 0.0625 | 1,6384 |
| 46R | Negative | 16 | 1,024 | 0.015 | 65,536 |
| 53R | Negative | 8 | 1,024 | 0.0625 | 16,384 |
| SH73 | Positive | 32 | 1,024 | 0.25 | 8,192 |
| F232 | Positive | 32 | 1,024 | 0.0312 | 32,768 |
| YLG | Negative | 32 | 1,024 | 5.0 | 2,048 |
| LEG | Negative | 32 | 1,024 | 0.0312 | 32,768 |
| SH05 | Positive | 32 | 1,024 | 0.25 | 4,096 |
| S506 | Negative | 32 | 1,024 | <0.0156 | >65,536 |
| SH22 | Positive | 32 | 1,024 | 0.25 | 4,096 |
| SH30 | Positive | 32 | 1,024 | 0.125 | 8,192 |
| SH13 | Positive | 32 | 1,024 | 0.125 | 8,192 |
| S2 | Positive | 32 | 1,024 | <0.0156 | >65,536 |
| S57 | Negative | 32 | 1,024 | 0.0156 | 65,536 |
| F30 | Positive | 32 | 1,024 | 0.125 | 8,192 |
| S28 | Positive | 32 | 1,024 | 0.125 | 8,192 |
| F162 | Positive | 8 | 1,024 | 0.25 | 4,096 |
| S60s | Negative | 4 | 1,024 | 0.0156 | 65,536 |

[a]COL, colistin; MRC, minimum resistance reversal concentration; NCL, niclosamide.

counts of *Salmonella* SH05 in the presence of niclosamide, in combination with colistin, decreased by 3.25 and 3.95 $\log_{10}$ CFU/mL, respectively, compared with colistin alone at 24 h of incubation (Fig. 1C). More importantly, the MIC of colistin in the presence of 1 or 4 µg/mL niclosamide after 30 serial passages remained below the susceptibility breakpoint. Indeed, our data revealed that the presence of low concentrations of niclosamide suppressed the development of colistin resistance (Fig. 1D). Furthermore, at the 1 µg/mL concentration of niclosamide, only the MICs of florfenicol for *Salmonella* SH05 decreased twofold. Moreover, when increasing the niclosamide concentration to 4 µg/mL, only the MIC of doxycycline and florfenicol decreased twofold for *Salmonella* SH05 (Fig. 1E). Specifically, niclosamide, as an adjuvant of antibiotics, showed synergism with colistin but not with other antibiotics. Altogether, our results established that niclosamide restored colistin sensitivity without affecting bacterial viability and without inducing colistin resistance.

## Dose effect of niclosamide on reversing colistin resistance

We evaluated the dose–response relationship of the reversal effect of different concentrations of niclosamide on colistin resistance using the broth microdilution susceptibility test. In the present study, commonly used indicators such as MIC, MRC, and PRC can only reflect the ability of niclosamide to reverse drug resistance but cannot comprehensively reflect the efficiency of its reversal of drug resistance. In view of this, we adopted the RRI to precisely evaluate the efficiency of niclosamide in reversing drug resistance, to provide more accurate and practical reference data for clinical practice. Interestingly, we found that there was a specific dose–response relationship in the reversal effect of niclosamide on colistin resistance, and it was not concentration dependent. The study found that when the concentration of niclosamide was within the range of MRC ~2 µg/mL, its RRI was ≥1, and the MIC of colistin was decreased by at least 16-fold, indicating that within this concentration range, niclosamide had the highest reversal efficiency against colistin resistance. At this point, the ratio of the MIC of colistin to the concentration of niclosamide was 1:8–16:1. In addition, although a higher concentration

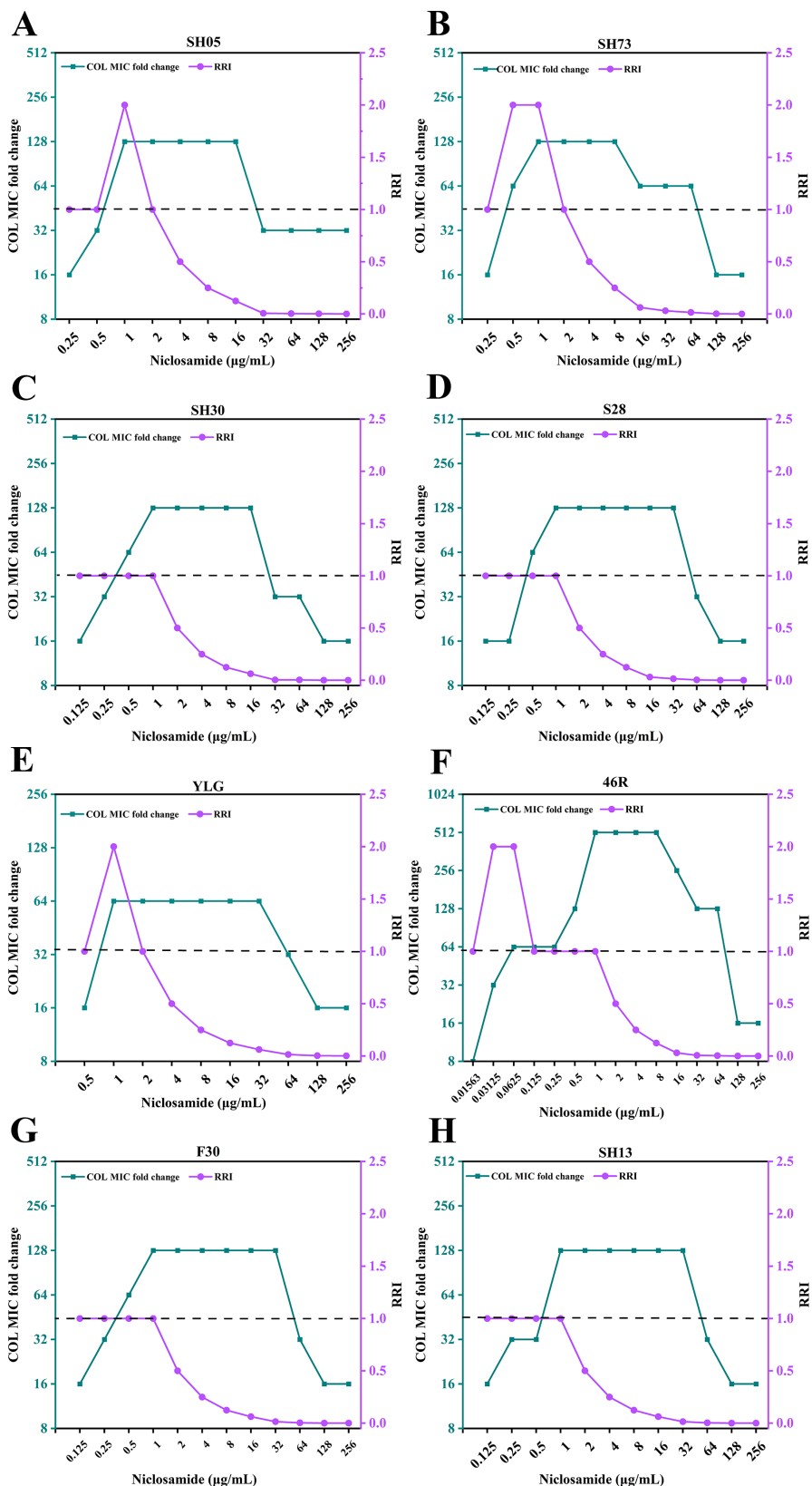

**FIG 2** Dose–response curves between niclosamide and its reversal effect on colistin resistance. COL, colistin; RRI, resistance reverse index. Curves were generated with niclosamide concentration (x-axis), RRI (right y-axis), and COL MIC fold change (left y-axis); (A–H) represent strains SH05, SH73, S28, SH30, 46R, YLG, F30, and SH13, respectively.

of niclosamide (4–256 µg/mL) reduced the MIC of colistin by at least eightfold, the RRI gradually decreased, and its reversal efficiency also decreased accordingly (Fig. 2).

Notably, the PRCs of niclosamide for all strains include multiple concentrations of niclosamide, and within this concentration range, niclosamide demonstrated a strong ability to reverse colistin resistance. Within the range of the PRC, as the concentration of niclosamide gradually increased, although the ability to reverse colistin resistance remained unchanged, the RRI gradually decreased, and the reversal efficiency of colistin resistance also gradually declined (Table S3). Overall, compared with a high concentration of niclosamide, a low concentration range of niclosamide had the highest reversal efficiency against colistin resistance and exhibited higher safety and effectiveness in clinical applications. These studies were of great significance for determining the dosage of niclosamide in clinical use and the ratio of the two drugs. Specifically, a possible method to improve the water solubility and bioavailability of niclosamide was to develop a preparation (through encapsulation). According to the dose–response relationship in this study, the developed preparation was able to effectively and selectively mix niclosamide and polymyxin in various ratios ranging from 1:8 to 16:1 and directly deliver them to the pathogens at the site of infection. Qin et al. (29) prepared negatively charged poly(ethylene glycol)-functionalized liposomes (Lipo-cc) encapsulating both curcumin and colistin for the co-delivery of these two drugs and successfully overcame colistin resistance.

In conclusion, in the clinical combination therapy for the treatment of infections caused by multidrug-resistant bacteria, higher doses of the adjuvant do not necessarily lead to higher efficiency in reversing colistin resistance. Therefore, the adjuvant with the highest efficiency in reversing colistin resistance (RRI ≥1) and at a lower concentration is more likely to be applied in clinical treatment. The introduction of the RRI in this study provided a scientific theoretical basis for determining the optimal dose range of the adjuvant in the clinical combination medication regimen or the dose range of its compound preparation.

## Mechanism study on the enhancement of colistin antibacterial efficacy by niclosamide

The OM of gram-negative bacteria is an effective barrier in the outer leaf that protects bacteria from toxic environmental insults. Colistin binding to the negatively charged lipid A phosphate groups of an LPS component causes membrane destabilization, leading to an increase in cell envelope permeability, leakage of cellular contents, and, ultimately, cell death (30). The mechanism of colistin resistance in gram-negative bacteria mainly involves the modification of lipid A, rendering the binding of colistin ineffective (31). Therefore, we hypothesized that the addition of niclosamide would restore the ability of colistin to disrupt the bacterial membrane. To verify this hypothesis, we first used scanning electron microscopy to observe the morphological changes in *Salmonella* SH05 treated with colistin or niclosamide alone or in combination. Compared with the single treatment, the bacteria in the combination group were significantly damaged, and cytoplasmic leakage could be observed, accompanied by obvious damage to the OM (Fig. 3A). To further confirm this theory, a live/dead bacterial staining analysis was employed to evaluate the membrane permeability of colistin-resistant *Salmonella* before and after treatment with the colistin and niclosamide combination. Live/dead staining further demonstrated that the treatment with colistin–niclosamide combination caused significant damage to the bacterial membrane, leading to bacterial lysis (Fig. 3B). The envelope of bacterial cells is a chemical compartment that helps maintain cellular homeostasis and serves as a barrier to provide selective permeability for various substances (32). Subsequently, we tested the effect of niclosamide or colistin alone or in combination on the permeability of the OM by fluorescence intensity analysis. We found that compared with colistin or niclosamide alone, the combination of colistin (1/2× MIC) and niclosamide (1 µg/mL) dramatically increased the fluorescence intensity of NPN (Fig. 3C), indicating that their combination observably improved the permeability

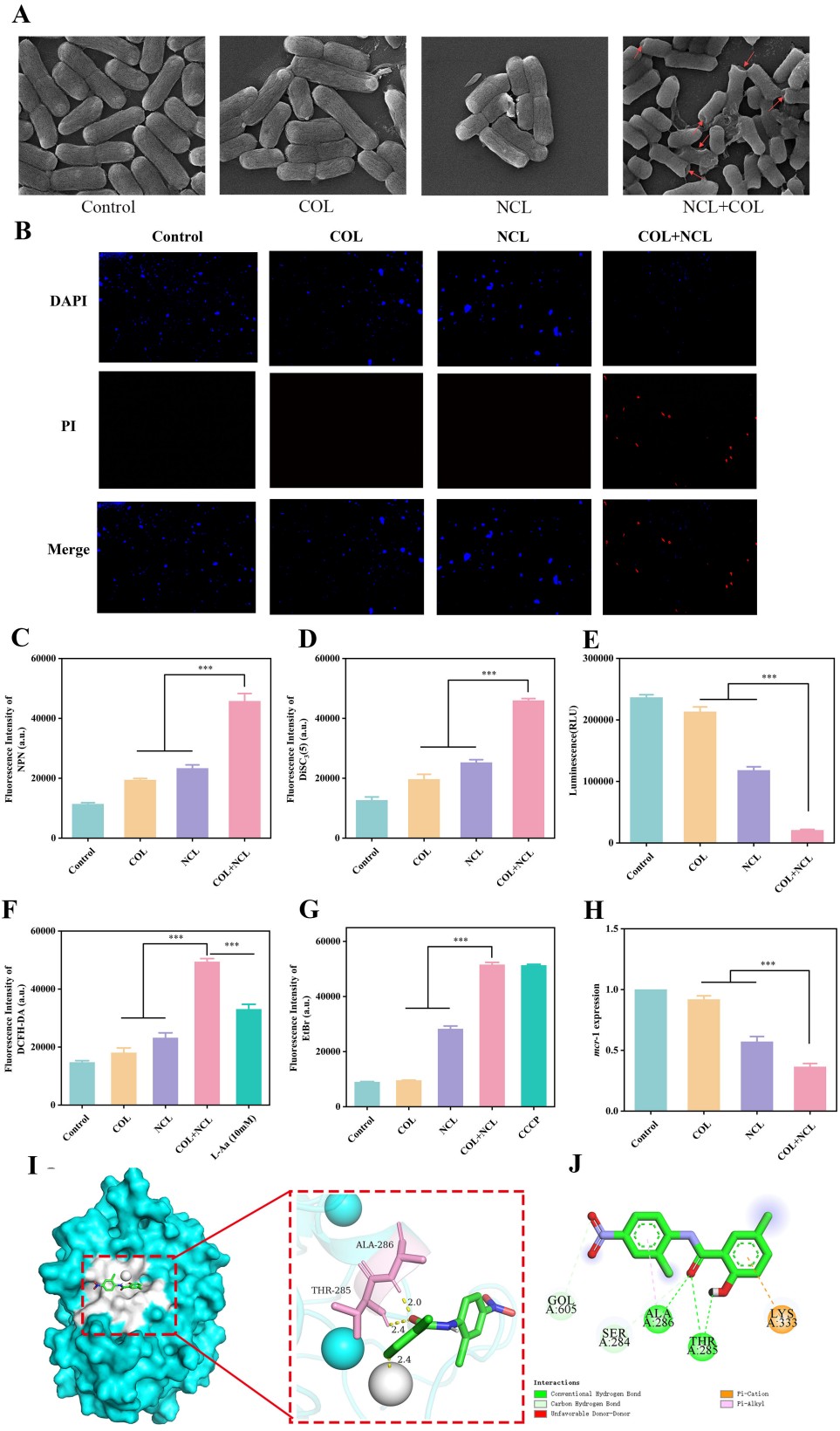

**FIG 3** Synergistic mechanisms of niclosamide–colistin combination. (A) Morphological changes in *Salmonella* SH05 treated with niclosamide (1 μg/mL), colistin (1/2× MIC), or their combination were observed by transmission electron microscopy. (B) Live/dead staining of *Salmonella* SH05 was performed after treatment with niclosamide (1 μg/mL), colistin (1/2× MIC), (Continued on next page)

**Fig 3 (Continued)**

or niclosamide (1 µg/mL) plus colistin (1/2× MIC). Viable bacterial cells were stained blue by 4′,6-diamidino-2-phenylindole (DAPI), and dead bacterial cells were stained red by propidium iodide. The scale bar is 300 µm. (C) Membrane permeability analysis of bacteria treated with niclosamide (1 µg/mL), colistin (1/2× MIC), or niclosamide plus colistin via fluorescence of NPN. (D) The fluorescence intensities of $DiSC_3(5)$ were measured after treatment with colistin alone and various concentrations of niclosamide combined with colistin. (E) Decreased levels of intracellular ATP in *Salmonella* SH05 upon treatment with niclosamide–colistin combination. (F) ROS levels were determined after treatment with colistin or colistin combined with niclosamide in different concentrations. Exogenous addition of ROS quencher L-ascorbic acid (10 mM) diminished the accumulation of ROS induced by the combination. (G) The accumulation of EtBr was used to assess the activity of efflux pumps induced by niclosamide, colistin, or their combinations. (H) Niclosamide (1 µg/mL), in combination with colistin (1/2× MIC), efficiently reduced expression of the *mcr*-1 gene. (I) Prediction of binding modes of niclosamide with MCR-1 protein using molecular modeling. (J) Interaction of planar amino acid residues between niclosamide and the MCR-1 molecule. All data are presented as mean ± SD, and the significances were determined by non-parametric one-way analysis of variance (***$P < 0.001$). DCFH-DA, 2′,7′-dichlorodihydrofluorescein diacetate.

of the OM. As expected, treatment with the combination of niclosamide and colistin led to a drastic increase in membrane permeability, indicating that the effective bactericidal concentration of colistin could be reduced in the presence of niclosamide. Together, these findings indicated that niclosamide restored colistin activity by potentiating the membrane-damaging ability of colistin.

Since the increase in membrane permeability usually causes the dissipation of membrane potential, we speculated that niclosamide might disrupt *Salmonella* proton motive force (PMF). PMF is the electrochemical gradient of protons generated by the electron transport chain in bacteria, and its function is to expel protons out of the cell (33). To test this, $DiSC_3(5)$ was used to evaluate the bacterial membrane potential. We found that the combined use of niclosamide (1 µg/mL) and colistin (1/2× MIC) resulted in enhanced fluorescence intensity, while the use of colistin alone had a weaker effect on the loss of membrane potential, indicating that niclosamide disrupted the Δψ of *Salmonella* isolates (Fig. 3D). This indicates that niclosamide can affect bacterial PMF by affecting the dissipation of membrane potential. Because PMF is the driving force for ATP synthesis, the change in the ATP level will affect the function of cells, and it plays an important role in various physiological processes of cells (33). We further evaluated the intracellular ATP levels of *Salmonella* treated with the combination of niclosamide and colistin using the ATP Assay Kit. We found that the luminescence intensity of *Salmonella* treated with a combination of niclosamide and colistin was significantly reduced compared to colistin alone, which was consistent with the dissipation of PMF (Fig. 3E). Previous studies demonstrated that membrane depolarization is related to the production of ROS and PMF (10). A new report also shows that the oxidative stress due to polymyxin-induced formation of ROS induces rapid cell death through the accumulation of hydroxyl radical (•OH). Therefore, we hypothesized that the combination of niclosamide and colistin may enhance oxidative damage. A high level of ROS induces lipid peroxidation of the cell membrane and further damages biological macromolecules, such as proteins and DNA, which eventually results in the death of pathogenic bacteria (34). As expected, combination therapy with niclosamide and colistin led to a significant increase in oxidative damage to bacteria by inducing excessive generation of ROS (Fig. 3F). ROS plays an important role in the bactericidal processes of antibiotics (35). Therefore, we speculate that niclosamide induces excessive ROS production, which may interfere with various molecules such as lipids, cytoplasmic proteins, and DNA, ultimately facilitating the killing of bacteria by colistin. When the ROS scavenger L-Aa (10 mM) was added to the sample with niclosamide and colistin, the accumulation of ROS was diminished (Fig. 3F). In fact, the supplement of L-Aa increased the bacterial survival rate following the treatment of niclosamide and colistin (Fig. S2). Thus, niclosamide triggers the accumulation of ROS, which correspondingly aggravates membrane damage to further increase the antibacterial activity of colistin. These results indicate that the

combination of niclosamide and colistin can dissipate the PMF, reduce ATP production, and increase oxidative stress to cause cell death.

Second, considering that PMF is critical for the functions of efflux pumps, the effect of the combination of niclosamide and colistin on efflux pump activity was determined using the EtBr efflux assay (36). The fluorescence intensity of cells treated with niclosamide (1 µg/mL) alone significantly increased, indicating that EtBr accumulated in the cells because niclosamide inhibited the efflux pump. Moreover, the change in fluorescence intensity of niclosamide combined with colistin was considerably higher than that of cells treated with colistin or niclosamide alone, with values comparable to those of the positive control CCCP, which is an efflux pump inhibitor (Fig. 3G). The results indicated that the combined use of niclosamide and colistin exerted an even stronger inhibitory effect on EtBr efflux, confirming that niclosamide and colistin act synergistically in inhibiting EtBr efflux.

Next, in order to explore the potential binding mode between niclosamide and MCR-1, molecular docking was performed. As shown in Fig. 3I and J, computational insights from molecular docking analysis revealed that niclosamide could bind to the groove on the surface of the MCR-1 protein and form hydrogen bonds with Thr285 and Ala286, respectively, in the form of a hydrogen bond. The findings revealed that the free energy of binding was −5.3 kcal/mol, suggesting that the MCR-1 protein and niclosamide potentially bind. An earlier study has demonstrated that zinc ions and amino acids such as Thr285, Glu246, His395, Asp465, His466, and His478 are very important for the biological activity of MCR-1 protein, and the region formed by them is considered to be the active site of MCR-1 protein (37). Our results indicate that niclosamide fits quite reasonably into the binding pocket of the MCR-1 protein, further interfering with the function of the MCR-1 protein. Moreover, the catalytic amino acid Thr285 in the active pocket of the MCR-1 protein was found to form a hydrogen bond with the phenolic hydroxyl group of niclosamide, thus stabilizing the docking structure. More importantly, results from the Western blot assay showed that the expression of MCR-1 in the MCR-1-positive bacterial strain *Salmonella* SH05 was not visibly affected by niclosamide treatment for 6 h, suggesting that niclosamide may inhibit MCR-1 activity by direct engagement (Fig. S3). In addition, the *mcr*-1 expression in *Salmonella* SH05 was downregulated in the presence of niclosamide (Fig. 3H). In other words, niclosamide might be a potential inhibitor of MCR-1 to rescue colistin activity against resistant strains.

In the mechanistic study, it was found that niclosamide acted synergistically with colistin to increase the permeability of the bacterial cell membrane, dissipate the PMF, inhibit the function of the multidrug efflux pump, reduce ATP production, enhance ROS stress, and inhibit the activity of MCR-1. The increase in membrane permeability caused by the combination of niclosamide and colistin allowed the self-promoted absorption of colistin molecules, further enhancing its destructive effect. The dissipation of PMF after treatment with niclosamide and colistin was associated with membrane depolarization and dysfunction. Since PMF is essential for driving efflux activity, the dissipation of PMF led to the inhibition of drug efflux and the rapid accumulation of drugs within the bacterial cells. Meanwhile, niclosamide also inhibited the activity of the efflux pump, resulting in further cellular damage. In addition, the combined treatment with niclosamide and colistin significantly increased the oxidative damage of bacteria by inducing the excessive production of ROS, thus enhancing the antibacterial effect of the combination (Fig. 4).

## Preparation and characteristics of NCL@mPEG-PLGA-NPs nanoparticles

To overcome the poor solubility and low absorption of niclosamide, we designed a drug delivery system to enhance its bioavailability and improve the therapeutic effect of colistin against resistant *Salmonella*. The response surface methodology showed that the optimal values of NCL@mPEG-PLGA-NP EE (85.17%), drug LE (4.56%), and particle size (size = 127.1 nm) occurred at a mPEG-PLGA concentration of 4.99%, a 1:1 ratio of acetone to dichloromethane, and a PVA concentration of 1% (Fig. S4). Given

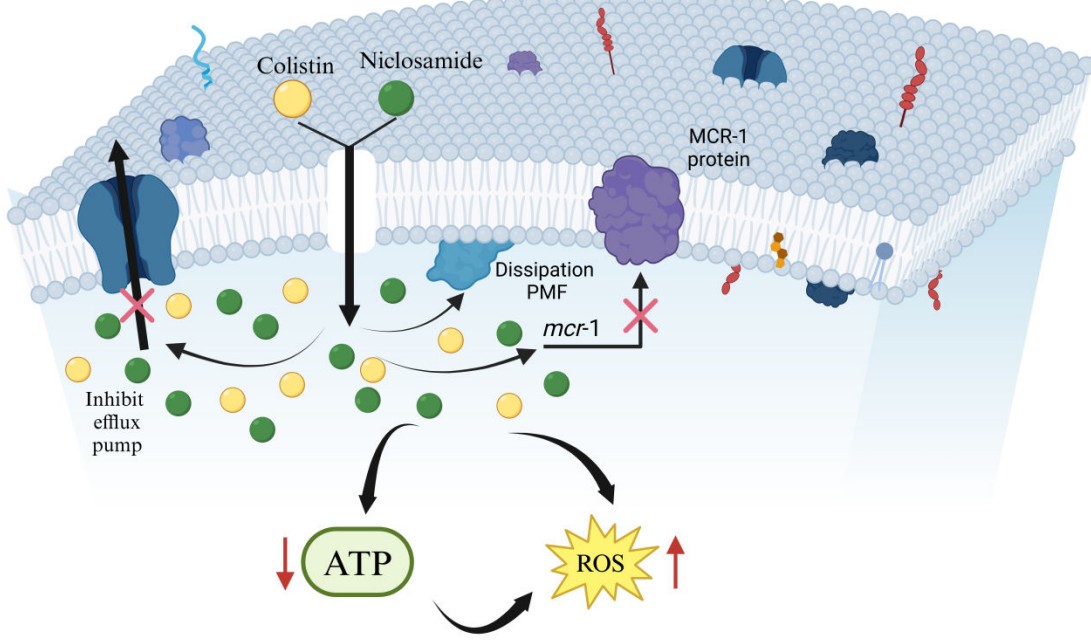

**FIG 4** Proposed antimicrobial mechanisms of the niclosamide and colistin combination on colistin-resistant *Salmonella*. The synergistic antibacterial activity of niclosamide and colistin involves increasing bacterial membrane permeability, dissipating proton motive force, and inhibiting efflux pumps. This leads to membrane damage, cytoplasmic leakage, inhibition of ATP production, and enhancement of ROS-mediated oxidative stress, ultimately resulting in bacterial cell death.

this, NCL@mPEG-PLGA-NPs were obtained, and their morphology, viewed by scanning electron microscopy, presented a smooth, rounded, and unbroken surface (Fig. 5A). The average particle size of NCL@mPEG-PLGA-NPs was 127.1 ± 2.18 nm; PDI was 0.182 ± 0.03; and zeta potential was −5.03 ± 0.64 mV (Fig. 5B and C). The relatively neutral surface charge (−5.03 ± 0.64 mV) of the PEGylated nanoparticle was advantageous for stability in a dynamic environment. In addition, Fourier transform infrared spectroscopy results showed that in the NCL spectrum, the phenolic hydroxyl group had a peak at 3,577.16 cm$^{-1}$; there was a stretching vibration peak of C-H on the benzene ring at 3,095.67 cm$^{-1}$; the peak at 1,652.80 cm$^{-1}$ represented C = O of the amide group. In the spectrum of mPEG-PLGA, the characteristic peak of mPEG-PLGA was seen at 1,746.14 and 1,080.30 cm$^{-1}$. In the spectrum of NCL@mPEG-PLGA-NPs (Fig. 5D), the characteristic absorption peaks of NCL and mPEG-PLGA appeared to change little, indicating that the basic skeleton of the nanoparticle did not change greatly. Serum stability is a key parameter for evaluating whether a targeted drug delivery system is suitable for *in vivo* applications. As shown in Fig. 5E, after incubation with 50% serum in PBS at 37°C for 3 days, the size of NCL@mPEG-PLGA-NPs did not change significantly, indicating that it had good size stability. Moreover, the size and zeta potential of NCL@mPEG-PLGA-NPs dispersed under different aqueous media remained stable, including fasted state simulated gastric fluid and intestinal fluid, and no significant aggregates were observed over 3 days (Fig. 5F and G). The particle size, PDI, and zeta potential of NCL@mPEG-PLGA-NPs at 4°C increased slowly within 30 days, but the differences were not significant, suggesting that the NCL@mPEG-PLGA-NPs had good long-term storage stability (Fig. 5H and Fig. S5). The research results showed that the nanoparticles could maintain stability in different media, indicating that the polyethylene glycolated nanoparticles had excellent stability and could provide prolonged blood circulation, which was ideal for the treatment of bacterial infections. The release rate of NCL@mPEG-PLGA-NPs reached nearly 31.61% within 6 h, reached approximately 66.21% at 24 h, and the release rate

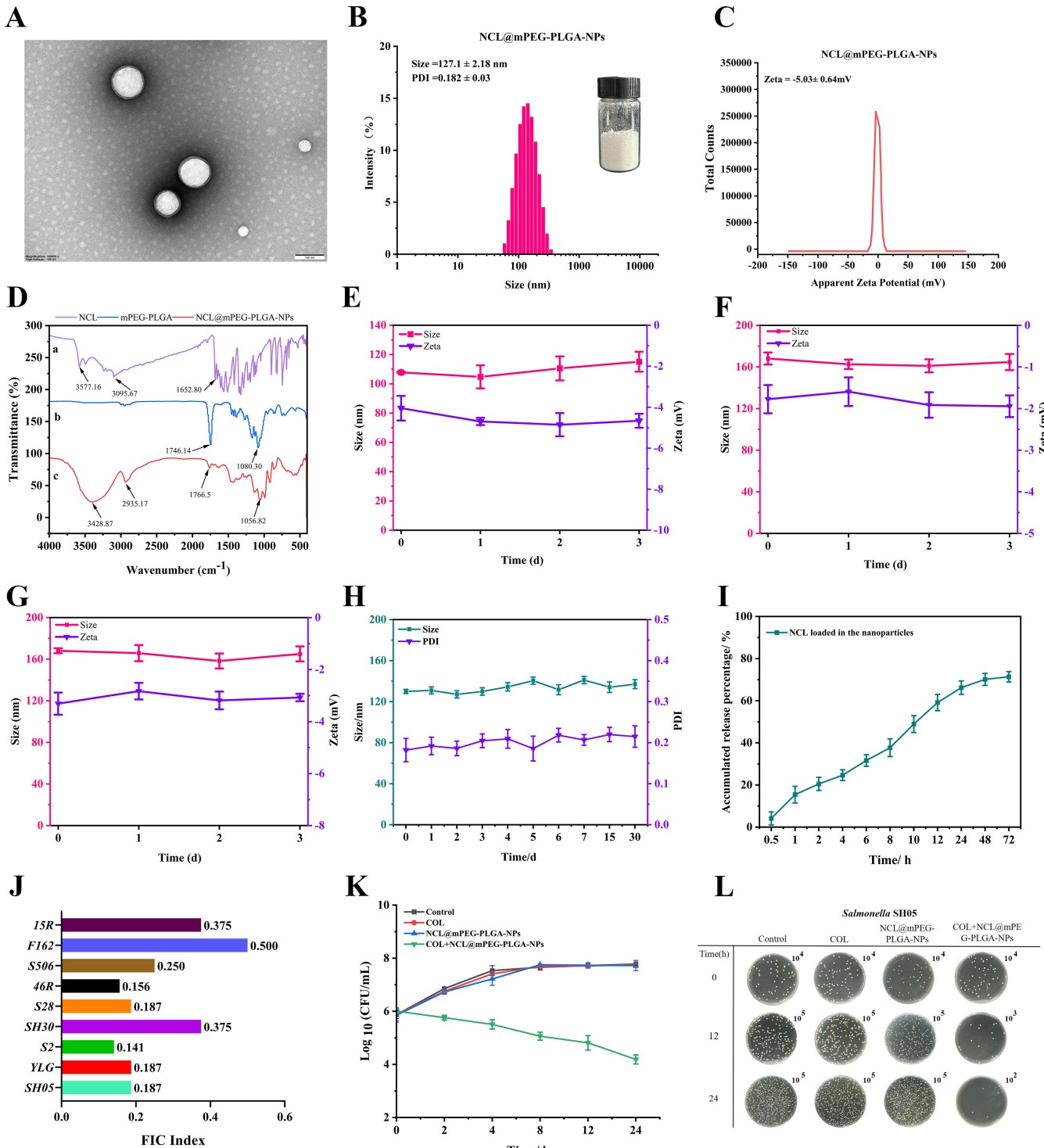

**FIG 5** Preparation and characterization of NCL@mPEG-PLGA-NPs. (A) Transmission electron microscopy image of NCL@mPEG-PLGA-NPs. (B and C) Mean particle size and zeta potential of NCL@mPEG-PLGA-NPs were measured. (D) Fourier transform infrared spectra of niclosamide, mPEG-PLGA, and NCL@mPEG-PLGA-NPs. (E) Stability assessment of NCL@mPEG-PLGA-NPs in 50% serum at 37°C. (F and G) Changes in the size and zeta potential of NCL@mPEG-PLGA-NPs incubated in various media at 37°C for 3 days (*n* = 3). (H) Changes in the size and PDI of NCL@mPEG-PLGA-NPs stored in PBS buffer at 4°C for 30 days. (I) *In vitro* cumulative release profiles of niclosamide released from NCL@mPEG-PLGA-NPs at 37°C in PBS. (J) Synergistic activity of NCL@mPEG-PLGA-NPs with colistin (FICI) against colistin-resistant *Salmonella* strains. (K) Time–kill curve of *Salmonella* SH05 treated with the combination of NCL@mPEG-PLGA-NPs (64 µg/mL) and colistin (2 µg/mL). (L) Colony images of colistin-resistant *Salmonella* SH05 treated with various formulations for 24 h at 37°C.

became slower, reaching 71.36% at 72 h (Fig. 5I). The release profiles indicated that niclosamide was released completely from nanoparticles within 72 h.

More importantly, NCL@mPEG-PLGA-NPs were found to significantly synergize with colistin against all the tested resistant strains using a checkerboard assay, with an FICI of 0.141–0.5 (Fig. 5J). As shown in Fig. 5K, neither NCL@mPEG-PLGA-NPs nor colistin monotreatment killed exponentially growing *Salmonella*. In contrast, Fig. 5K and L show that 64 µg/mL NCL@mPEG-PLGA-NPs (NCL: 0.47 µg/mL) in combination with 2 µg/mL colistin demonstrated higher synergistic activity at 24 h, leading to a greater reduction in the bacterial counts. The bacterial cell counts of *Salmonella* SH05 in the presence of NCL@mPEG-PLGA-NPs in combination with colistin decreased by 3.53 $\log_{10}$ CFU/mL, compared with colistin alone at 24 h of incubation. These results demonstrated the significant synergistic antibacterial effects of NCL@mPEG-PLGA-NPs and colistin, which may provide a new strategy for treating infections caused by resistant bacteria. In particular, niclosamide is encapsulated in polymer mPEG-PLGA, which has sustained-release properties, enabling NCL@mPEG-PLGA-NPs to be used in combination with colistin to provide long-lasting bactericidal effects.

## Safety evaluation of NCL@mPEG-PLGA-NPs

A key factor limiting the clinical application of adjuvants is their potential toxic effects. To evaluate the toxicity of niclosamide, we analyzed the hemolysis caused by niclosamide and NCL@mPEG-PLGA-NPs. It is encouraging that we did not observe any enhanced toxicity of NCL@mPEG-PLGA-NPs. On the contrary, we found that the hemolysis rate of low concentration niclosamide (4–64 µg/mL) was less than 5% (Fig. 6A and B). In addition, we evaluated the impact of niclosamide on liver and kidney function through biochemical indicators. As shown in Fig. 6D through F, compared with the colistin group, the levels of alanine aminotransferase, urea, and creatinine in the groups treated with COL + NCL and COL + NCL@mPEG-PLGA-NPs were significantly reduced ($P < 0.01$), indicating that niclosamide can alleviate liver and kidney damage in mice within a certain concentration range. Therefore, NCL@mPEG-PLGA-NPs have good biocompatibility both *in vitro* and *in vivo*.

## *In vivo* synergistic effect of NCL@mPEG-PLGA-NPs with colistin against *Salmonella* infection

Given that the combination of colistin and NCL@mPEG-PLGA-NPs exhibits significant synergistic bactericidal activity against *Salmonella in vitro*, we subsequently evaluated whether these effects could yield positive outcomes in animal infection models. To confirm this, we tested the *in vivo* efficacy of the combination of colistin (5 mg/kg) and NCL@mPEG-PLGA-NPs (20 and 40 mg/kg of NCL@mPEG-PLGA-NPs, which contain equivalent doses of 5.48 and 10.96 mg/kg of niclosamide, respectively) in infection models infected with resistant *Salmonella* SH05. As shown in Fig. 7B, infected mice treated with the combination of colistin and NCL@mPEG-PLGA-NPs had a higher survival rate (62.5%). Most notably, the majority of mice survived within 7 days after treatment with a single dose of colistin combined with NCL@mPEG-PLGA-NPs (20 and 40 mg/kg), outperforming the group treated with colistin plus niclosamide (5 + 10 mg/kg). These results indicated that the combination of colistin and NCL@mPEG-PLGA-NPs showed an effective protective effect against *Salmonella* infections. In addition, the combination therapy of colistin and NCL@mPEG-PLGA-NPs (5 + 20 mg/kg) reduced the bacterial load in the liver and spleen of mice by 3.84 and 3.29 $\log_{10}$ CFU/g, respectively, when compared with colistin monotherapy. Encouragingly, compared with colistin plus free niclosamide, the number of colonies in the liver and spleen of mice in the colistin plus NCL@mPEG-PLGA-NPs group was significantly decreased by 1.82–2.92 $\log_{10}$ CFU/g ($P < 0.001$) (Fig. 7C and D). Importantly, compared with the mice that were untreated or treated with niclosamide, colistin, or the combination of niclosamide and colistin, the levels of pro-inflammatory cytokines interleukin-6 (Fig. 7E) and tumor necrosis factor alpha (Fig. 7F) in the infected mice after treatment with NCL@mPEG-PLGA-NPs

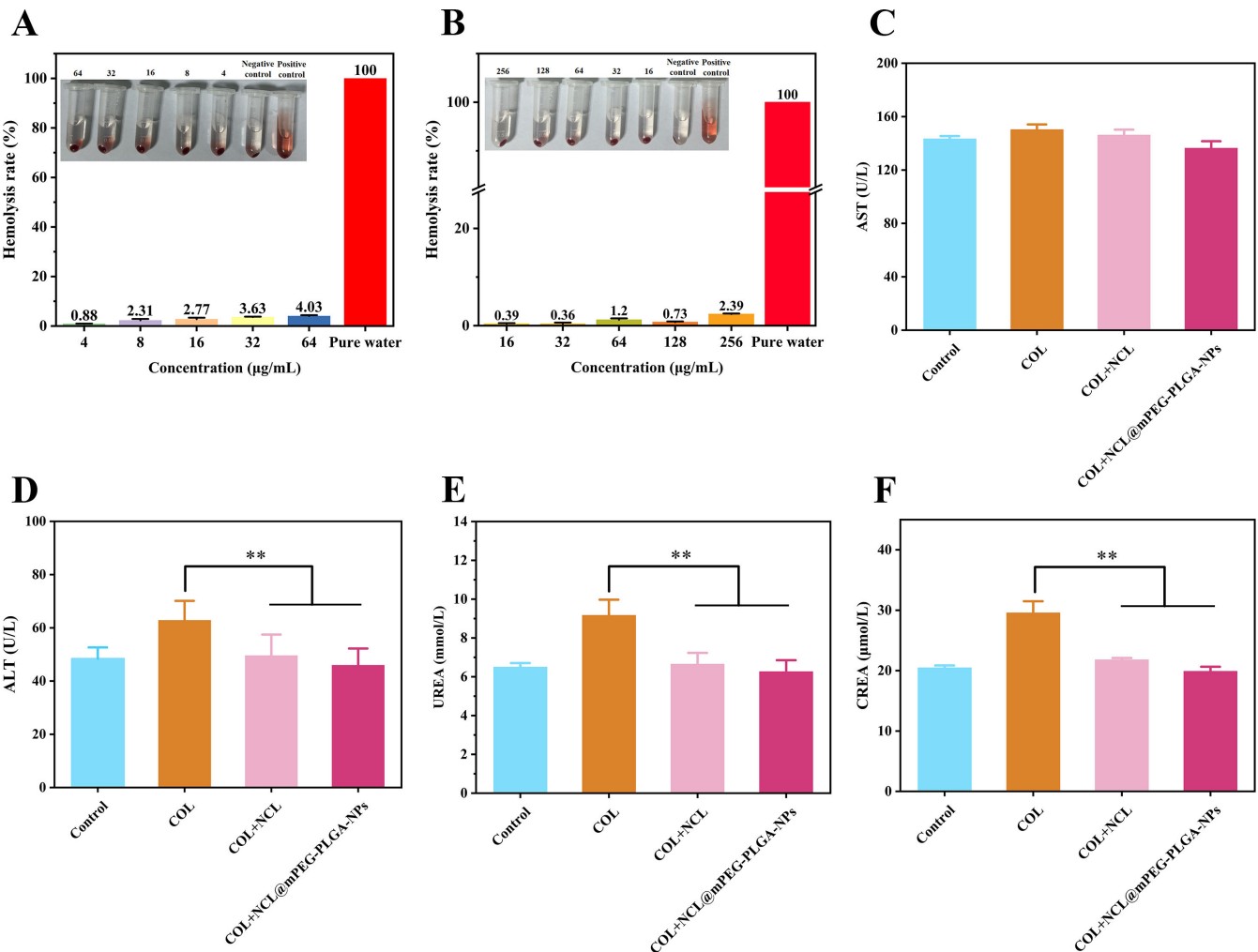

**FIG 6** Biosafety evaluation of niclosamide and NCL@mPEG-PLGA-NPs *in vitro* and *in vivo*. (A and B) Hemolysis evaluation of niclosamide and NCL@mPEG-PLGA-NPs was performed. (C–F) Liver and renal function-related indices (aspartate aminotransferase [AST], alanine aminotransferase [ALT], urea [UREA], and creatinine [CREA]) in mice after administration of niclosamide and NCL@mPEG-PLGA-NPs were analyzed. All data are presented as mean ± SD, and significances were determined by one-way analysis of variance (**$P < 0.01$).

plus colistin were significantly decreased. All these results indicated that the treatment regimens of the combination of colistin and NCL@mPEG-PLGA-NPs had a significant selective advantage in treating *Salmonella* infection.

*In vivo*, drugs need to reach the infected site in sufficient concentrations and exhibit therapeutic activity, which is inseparable from effective absorption and bioavailability. In this study, although the combination therapy of niclosamide and colistin showed synergistic antibacterial activity against colistin-resistant *Salmonella in vitro*, neither the combination therapy nor monotherapy had effective therapeutic effects on *Salmonella*-infected mice due to the low bioavailability and low blood concentration of niclosamide *in vivo*. Therefore, to address the issue of low bioavailability, we developed a nanodelivery system, NCL@mPEG-PLGA-NP, which is expected to significantly improve the *in vivo* absorption and bioavailability of niclosamide. We observed that the combination of NCL@mPEG-PLGA-NPs and colistin provided systemic protection against *Salmonella* infection, significantly increasing survival rates and reducing bacterial burden and inflammatory responses. Therefore, formulating drugs in the form of mPEG-PLGA nanoparticles is a key priority in pharmaceutics to enhance the absorption and bioavailability of niclosamide and further ensure its efficacy *in vivo*.

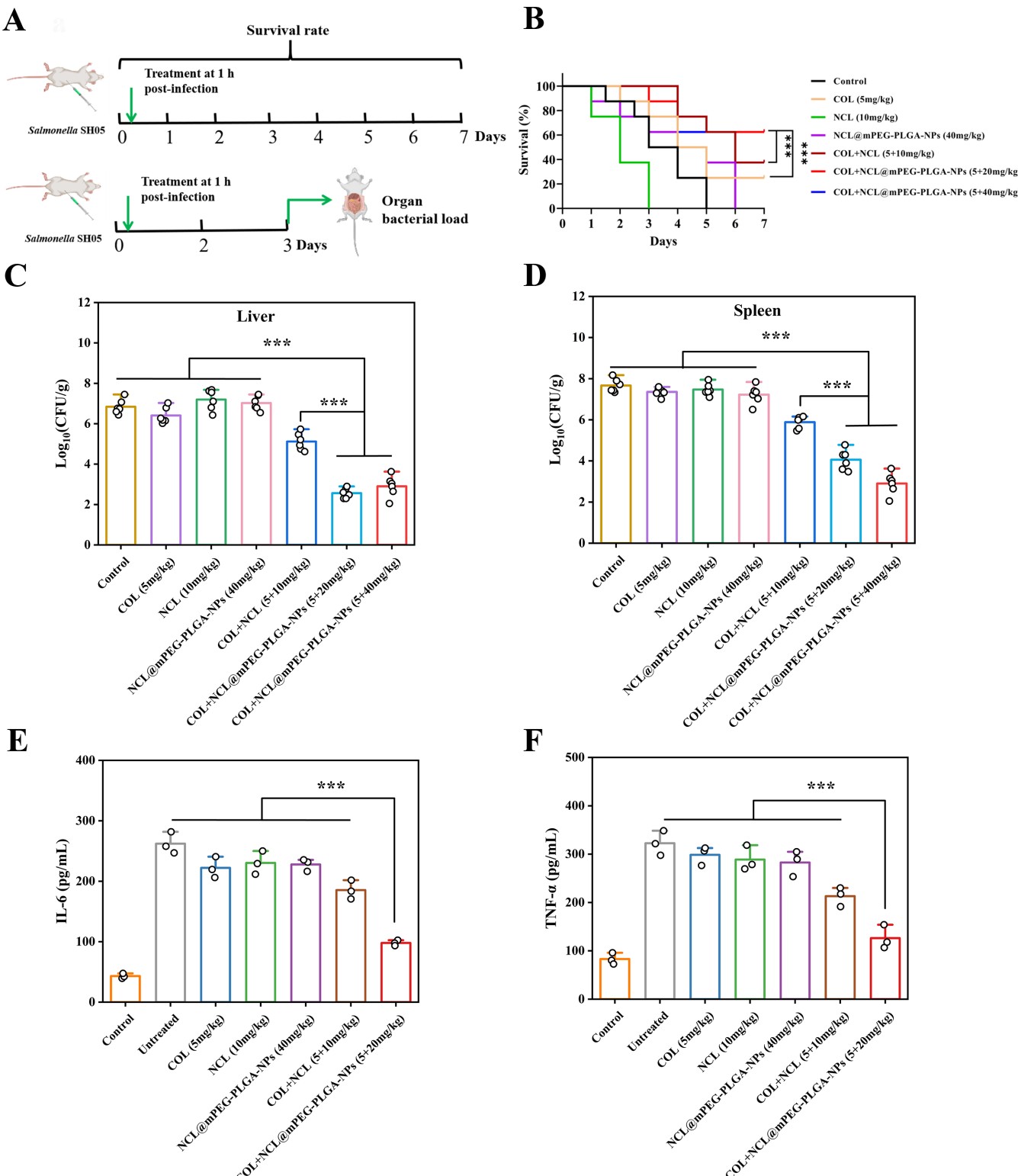

**FIG 7** The combination of NCL@mPEG-PLGA-NPs and colistin exhibited synergy *in vivo*. (A) Schematic of the experimental protocols for two animal infection models. (B) The combination of colistin (5 mg/kg) and NCL@mPEG-PLGA-NPs (20 and 40 mg/kg of NCL@mPEG-PLGA-NPs, which contain equivalent doses of 5.48 and 10.96 mg/kg of niclosamide, respectively) significantly improved the survival rate of female BALB/c mice (*n* = 6 per group) infected with colistin-resistant *Salmonella* SH05 (*mcr*-1 positive) compared with colistin monotherapy (5 mg/kg). (C and D) BALB/c mice were challenged with a sublethal dose of *Salmonella* SH05 (*mcr*-1 positive) and received a single intraperitoneal injection of vehicle, niclosamide, colistin, or a combination of colistin plus NCL@mPEG-PLGA-NPs (Continued on next page)

Fig 7 (Continued)

(5 + 20 mg/kg or 5 + 40 mg/kg) ($n$ = 6 per group). Bacterial burden in the liver and spleen was determined. Serum levels of interleukin (IL)-6 (E) and tumor necrosis factor alpha (TNF-α) (F) were measured via enzyme-linked immunosorbent assay ($n$ = 3). All data are presented as the means and SDs from at least two independent experiments. $P$ values were determined by Mann–Whitney $U$ test. (\*\*\*$P$ < 0.001).

## Conclusion

In conclusion, niclosamide, an antiparasitic drug, can be used as a potentiator of the antimicrobial activity of colistin. Our data strongly indicate that there is a specific dose–response relationship between niclosamide and its role in reversing colistin resistance, and this relationship is not concentration dependent. Niclosamide exhibits a high efficiency in reversing colistin resistance within the low concentration range of MRC ~2 µg/mL, but the reversal efficiency gradually decreases as the concentration of niclosamide increases. In-depth mechanistic analysis shows that low concentrations of niclosamide enhance colistin activity through multiple strategies, including enhancing the membrane-damaging ability of colistin, disrupting the functions of the PMF and efflux pumps, inhibiting the activity of mobilized colistin resistance genes, and accelerating reactive oxygen species-mediated oxidative damage. Additionally, the combination of NCL@mPEG-PLGA-NPs and colistin demonstrates significant synergistic antibacterial activity against colistin-resistant *Salmonella* and can provide a sustained bactericidal effect. Notably, the antibacterial efficacy of the combination of NCL@mPEG-PLGA-NPs and colistin can improve the survival rate of mice infected with *Salmonella* and reduce the bacterial load and inflammation levels. Furthermore, the nanoparticles have been proven to be non-toxic both *in vitro* and *in vivo*. Overall, the study of the dose–response relationship of niclosamide in reversing colistin resistance and the construction of the NCL@mPEG-PLGA-NPs nanodrug delivery system lay a scientific foundation for the clinical application of colistin adjuvants and the development of suitable drug delivery systems.

## ACKNOWLEDGMENTS

This work was supported by the National Natural Science Foundation of China (32373069) and the National Key Research and Development Program of China (2023YFD1800105).

G.Z.H. and X.Y.M. designed this project; K.F.Y., P.Y.L., and M.Y.Z. performed experiments and drafted this article. Q.G.L., Z.B.L., and M.J.F. helped with checkerboard and time–kill analysis; D.D.H. and L.Y. helped with the animal experiments; G.Z.H. designed the experiments, supervised the project, and wrote the manuscript.

## AUTHOR AFFILIATIONS

[1]Henan Agricultural University, Zhengzhou, China
[2]Ministry of Education Key Laboratory for Animal Pathogens and Biosafety, Zhengzhou, China
[3]Shangqiu Meilan Biological Engineering Co., Ltd, Shangqiu, Henan, China

## AUTHOR ORCIDs

Xiaoyuan Ma ⓘ http://orcid.org/0009-0000-0917-4531
Gongzheng Hu ⓘ http://orcid.org/0000-0002-7042-5771

## FUNDING

| Funder | Grant(s) | Author(s) |
| --- | --- | --- |
| National Natural Science Foundation of China | 32373069 | GongZheng HU |

| Funder | Grant(s) | Author(s) |
|---|---|---|
| National Basic Research Program of China (973 Program) | 2023YFD1800105 | GongZheng HU |

## AUTHOR CONTRIBUTIONS

Kaifang Yi, Conceptualization, Formal analysis | Peiyi Liu, Conceptualization, Data curation | Mengyao Zhang, Conceptualization | Qiange Liu, Data curation, Formal analysis | Mengjing Feng, Conceptualization, Data curation | Zibo Li, Conceptualization | Dandan He, Data curation, Investigation | Li Yuan, Data curation, Validation, Visualization | Xiaoyuan Ma, Conceptualization, Formal analysis, Investigation | Gongzheng Hu, Conceptualization, Data curation, Formal analysis, Funding acquisition, Visualization, Writing – original draft, Writing – review and editing

## ETHICS APPROVAL

All the *in vivo* experiments were approved by the Administration of Affairs Concerning Experimental Animals of the State Council of the People's Republic of China (approved on 14 November 1988). All animal experiments have been approved by the Animal Care and Use Committee of Henan Agricultural University, and we have adhered to all relevant ethical regulations concerning animal use.

## ADDITIONAL FILES

The following material is available online.

### Supplemental Material

**Figures S1 to S5 and Tables S1 to S3. (Spectrum02252-25-s0001.docx).** Niclosamide nanoparticles as a novel adjuvant reverse colistin resistance via multiple mechanisms against multidrug-resistant *Salmonella* infections.

### Open Peer Review

**PEER REVIEW HISTORY (review-history.pdf).** An accounting of the reviewer comments and feedback.

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
