## [Reviewer comments · Microbiology Spectrum]

Microbiology Spectrum

Niclosamide nanoparticles as a novel adjuvant reverse colistin resistance via multiple mechanisms against multidrug-resistant *Salmonella* infections

Kaifang Yi, Peiyi Liu, Mengyao Zhang, Qiange Liu, Mengjing Feng, Zibo Li, Dandan He, Li Yuan, Xiaoyuan Ma, and GongZheng HU

Corresponding Author(s): GongZheng HU, Henan Agricultural University

Review Timeline:

Submission Date:	July 25, 2025
Editorial Decision:	August 5, 2025
Revision Received:	August 13, 2025
Accepted:	August 19, 2025

Editor: Gregory Wiedman

Reviewer(s): The reviewers have opted to remain anonymous.

Transaction Report:

DOI: <https://doi.org/10.1128/spectrum.02252-25>

Re: Spectrum02252-25 (Niclosamide nanoparticles as a novel adjuvant reverse colistin resistance via multiple mechanisms against multidrug-resistant Salmonella infections)

Dear Prof. Gongzheng Hu:

Thank you for the privilege of reviewing your work. Below you will find my comments, instructions from the Spectrum editorial office, and the reviewer comments.

Revision Guidelines

Sincerely,
Gregory Wiedman
Editor
Microbiology Spectrum

Dear editor,

We sincerely appreciate the valuable comments and suggestions provided by the editors. This manuscript has been transferred from Antimicrobial Agents and Chemotherapy (manuscript number: AAC01059-25) to Microbiology Spectrum (manuscript number: Spectrum02252-25). In response to the specific comments from the editors, we have addressed each point individually. These revisions do not affect our interpretation of the research findings. We are pleased to submit the revised manuscript and believe that these improvements have enhanced the persuasiveness of our study. We look forward to your feedback and hope that the revised manuscript meets the academic standards of the journal.

Thank you once again for your understanding and support.

Response to the editor:

1. While I appreciate the amount of work that has been performed, relevant cytotoxicity data is missing and efficacy data in a more relevant animal model is needed.

Response: (1) To our knowledge, one of the limitations of niclosamide lies in its cytotoxicity. Even though niclosamide was found to not rupture red blood cells (hemolysis) at a high concentration of 32 g/ml, it was reported to elicit marked decreased cell viability (50% cytotoxic concentration \sim 0.25 g/ml) of kidney HEK 293T/17 and liver HepG2 eukaryotic cell lines at 0.125 g/ml (1). A possible solution to these limitations is to develop a formulation (via encapsulation) that can effectively and selectively deliver niclosamide (and colistin) to the pathogen at the site of infection and prevent it from binding to plasma proteins. In addition, Poly (lactic-co-glycolic acid) (PLGA) and methoxy poly (ethylene glycol) (mPEG) approved by the FDA have been extensively applied to prepare nanoparticles, and have shown good biocompatibility in vivo, with their degradation products of carbon dioxide and water being highly safe (2). Importantly, we designed and prepared a nanosystem capable of co-loading colistin and niclosamide with different physicochemical properties into mPEG-PLGA nanoparticles

(COL/NIC-mPEG-PLGA-NPs), and this nanosystem exhibits no cytotoxicity toward RAW 264.7 and PK-15 cells at effective therapeutic doses (3). In summary, the core-shell nanoparticles we prepared (NCL@mPEG-PLGA-NPs) do not induce cytotoxicity either.

(2) The in vivo efficacy, safety, and therapeutic mechanisms of niclosamide-loaded mPEG-PLGA nanoparticles (NCL@mPEG-PLGA-NPs) in combination with colistin were evaluated using a murine model of peritoneal sepsis induced by multidrug-resistant *Salmonella* SH05 (*mcr-1* positive).

Survival Rate:

The COL + NCL@mPEG-PLGA-NPs groups (20 mg/kg and 40 mg/kg) showed significantly improved survival (62.5%) over 7 days, outperforming the COL + NCL combination group and COL monotherapy.

Bacterial Burden in Tissues:

At 48 h post-infection, the COL + NCL@mPEG-PLGA-NPs (5 + 20 mg/kg) group exhibited a 3.84 log₁₀ CFU/g and 3.29 log₁₀ CFU/g reduction in bacterial load in the liver and spleen, respectively, compared to COL monotherapy.

Compared to the COL + NCL group, bacterial counts in the liver and spleen were further reduced by 1.82–2.92 log₁₀ CFU/g ($p < 0.001$).

Inflammatory Response:

Serum levels of pro-inflammatory cytokines (IL-6 and TNF- α) were significantly lower in the COL + NCL@mPEG-PLGA-NPs groups compared to other treatment groups, indicating attenuated systemic inflammation.

Safety Profile:

Biochemical analyses (ALT, UREA, CREA) revealed that COL + NCL@mPEG-PLGA-NPs reduced liver and kidney damage compared to COL monotherapy ($p < 0.01$), confirming good in vivo biocompatibility.

The combination of NCL@mPEG-PLGA-NPs and colistin significantly enhances therapeutic efficacy against multidrug-resistant *Salmonella* infections in vivo, as evidenced by improved survival, reduced bacterial burden, and alleviated inflammation, with favorable safety profiles. These findings support the potential of

NCL@mPEG-PLGA-NPs as a promising adjuvant for colistin in clinical settings.

References

1. Gooyit M, Janda KD. 2016. Reprofiled anthelmintics abate hypervirulent stationary-phase *Clostridium difficile*. *Sci Rep* 6:33642.
2. Danhier F, Ansorena E, Silva JM, Coco R, Le Breton A, Préat V. 2012. PLGA-based nanoparticles: an overview of biomedical applications. *J Control Release* 161:505-22.
3. Yi K, Wang X, Li P, Gao Y, He D, Pan Y, Ma X, Hu G, Zhai Y. 2025. Amphiphilic mPEG-PLGA copolymer nanoparticles co-delivering colistin and niclosamide to treat colistin-resistant Gram-negative bacteria infections. *Commun Biol* 8:673.

Thank you again for the insightful comments and suggestions about our original manuscript. We appreciate for Editors and Reviewers' warm work, and hope the correction will meet with approval. Once again, thank you very much for your comments and suggestions.

Yours sincerely,

Gongzheng Hu

E-mail address: yaolilab@126.com.

Re: Spectrum02252-25R1 (Niclosamide nanoparticles as a novel adjuvant reverse colistin resistance via multiple mechanisms against multidrug-resistant Salmonella infections)

Dear Mr. GongZheng HU:

Your manuscript has been accepted, and I am forwarding it to the ASM production staff for publication. Your paper will first be checked to make sure all elements meet the technical requirements. ASM staff will contact you if anything needs to be revised before copyediting and production can begin. Otherwise, you will be notified when your proofs are ready to be viewed.

Sincerely,
Gregory Wiedman
Editor
Microbiology Spectrum